# Ultrasound-Guided Coarse Needle Biopsy Diagnosed Isolated Hepatic Malignant Melanoma with Undetermined Origin in TB Patient: A Case Report

**DOI:** 10.3390/diagnostics13010042

**Published:** 2022-12-23

**Authors:** Kailing Chen, Yi Dong, Weibin Zhang, Hong Han, Feng Mao, Hui Zhang, Wenping Wang

**Affiliations:** 1Department of Ultrasound, Zhongshan Hospital, Fudan University, Shanghai 200032, China; 2Department of Ultrasound, Xinhua Hospital Affiliated to Shanghai Jiao Tong Universitity School of Medicine, Shanghai 200092, China

**Keywords:** hepatic malignant melanoma, ultrasound-guided percutaneous coarse needle biopsy, contrast-enhanced ultrasound, ^18^F-FDG-PET/CT

## Abstract

Isolated hepatic malignant melanoma with undetermined origin is relatively rare and the imaging findings vary significantly in published studies. In this report, we described an elderly male patient with pulmonary tuberculosis who was diagnosed with isolated hepatic malignant melanoma with undetermined origin by ultrasound-guided percutaneous coarse needle biopsy (US-CNB). The hepatic melanoma was detected accidentally on chest CT. On contrast-enhanced ultrasound (CEUS), it presented an enhancement pattern of fast washin and slow washout. However, on magnetic resonance imaging (CEMRI), it showed non-rim hyperenhancement in the arterial phase but hypointensity in the late phase, mimicking hepatocellular carcinoma. With inconsistent results, the patient underwent fluorine-18-fluro-2-deoxy-D-glucose-positron emission tomography/computed tomography (^18^F-FDG-PET/CT). The mass showed mild ^18^F-FDG uptake with SUVmax of 4.7, and hypermetabolic nodules were observed in the lung, chest wall, thoracic vertebra, and pelvis. Due to the advanced stage of the tumor, US-CNB was performed to acquire a pathological diagnosis. The immunohistochemical staining suggested malignant melanoma. Of note, no primary tumor was revealed. Finally, the patient refused systemic therapy and died from tumor progression seven months later. Hence, CEUS and CEMRI is fundamental in the diagnosis of hepatic melanoma, and PET-CT is helpful in clinical staging. For controversial results, US-CNB is required to establish the pathological diagnosis.

**Figure 1 diagnostics-13-00042-f001:**
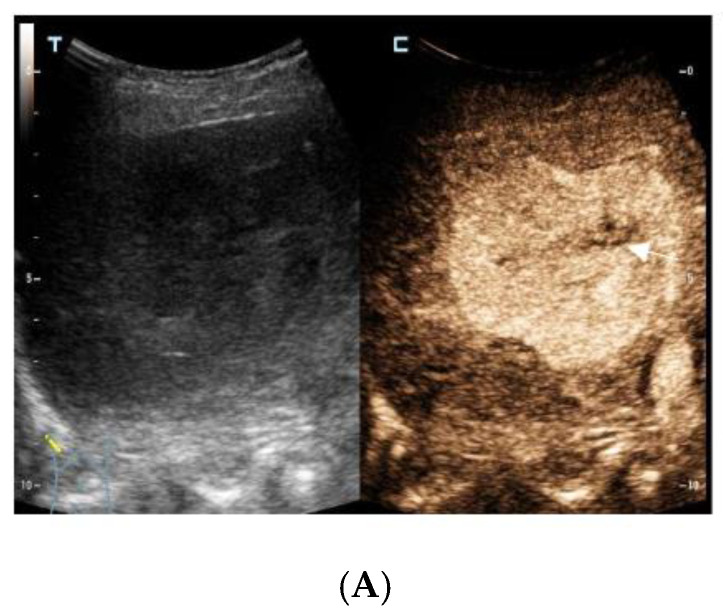
The contrast-enhanced ultrasound images of the hepatic lesion. A 71-year-old man presented at our medical center with intermittent fever in the morning and night for two weeks, accompanied by pain in the back. His body temperature was up to 39 °C, which subsided after anti-inflammatory treatment at the local hospital. He didn’t have a cough, expectoration, chest tightness, or joint pain. The medical history recorded that he had diabetes, hypertension, and suffered tuberculosis ten years ago. Currently, his spot test for mycobacterium tuberculosis (T-Spot) was negative, but his erythrocyte sedimentation rate (ESR) was 72 mm/H (normal range < 28 mm/H) and C-reaction protein (CRP) was 46.4 mg/L (normal range: 0–10 mg/L). Laboratory tests were negative for hepatitis B virus (HBV), hepatitis C virus (HCV), or human immunodeficiency virus (HIV) infection. The serum tumor markers of alpha-fetoprotein (AFP) and CYFRA 21-1 produced negative results, whereas the levels of carcino-embryonic antigen (CEA, 5.9 ng/mL, normal range: 0–5 ng/mL), carbohydrate antigen 199 (CA199, 45.0 U/mL, normal range: 0–34 U/mL), squamous cell carcinoma antigen (SCC, 3.8 ng/mL, normal range: 0–3.0 ng/mL), and neuron specific enolase (NSE, 16.4 ng/mL, normal range: 0–16.3 ng/mL) were slightly higher than the normal range. A computed tomography (CT) scan of the chest showed subcentimeter (2–4 mm) nodules in the lung, which were suspected to be tuberculosis or metastatic tumors. A low-density mass was accidentally observed in the right posterior lobe of the liver. On gray-scale ultrasound, a 65 mm hypoechoic lesion was revealed in segment seven (S7) of the liver with a regular shape and well-defined margin. After injection of SonVue^®^ (Bracco, Milan, Italy), the lesion showed heterogeneous hyperenhancement in the arterial phase (**A**). During portal venous and late phase, the lesion showed continuous isoenhancement compared with the surrounding liver parenchyma (**B**,**C**). It became slightly hypo-enhanced until seven minutes later (**D**). Unenhanced area was observed during the whole enhancement phase (**A**–**C**, arrow). Considering the medical history of fever, diabetes, and absence of HBV infection, the liver nodule was initially diagnosed as hepatapostema.

**Figure 2 diagnostics-13-00042-f002:**
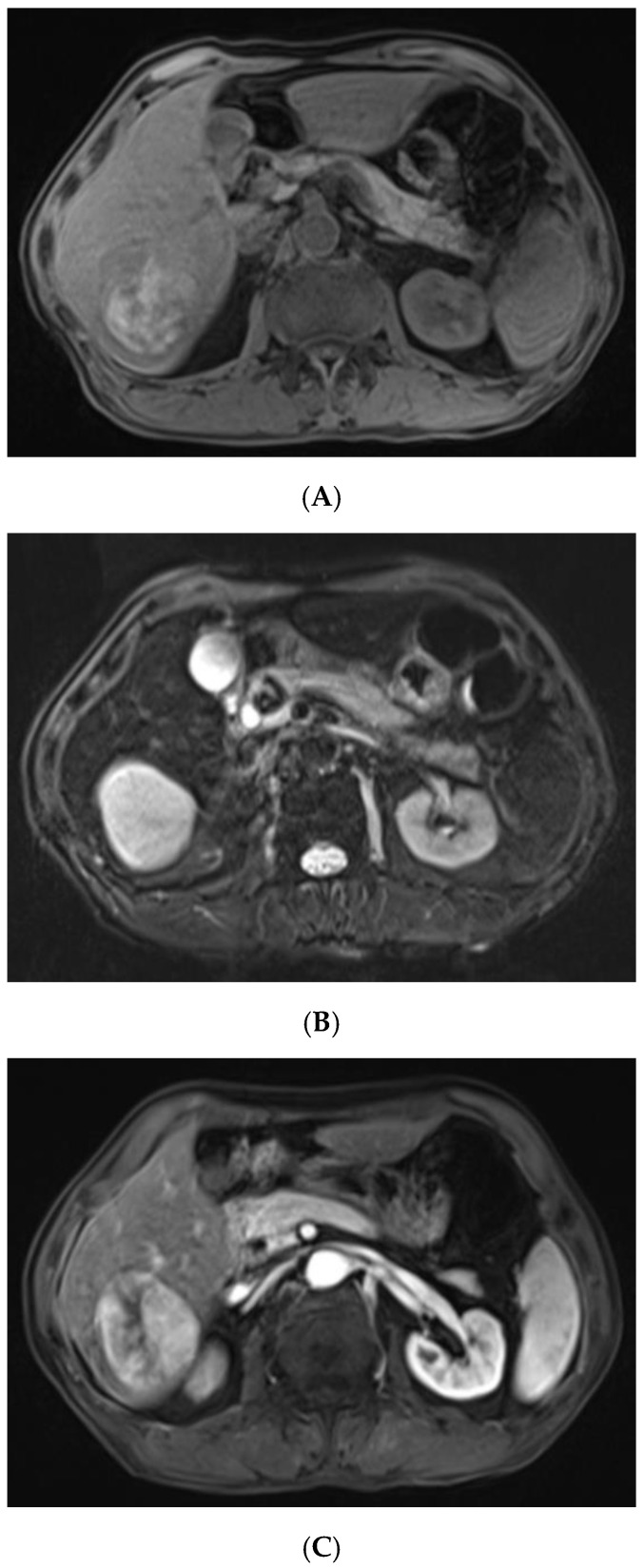
For further diagnosis, abdominal magnetic resonance imaging (MRI) was performed. The hepatic nodule was slight hyperintense on unenhanced T1WI (**A**). On T2WI, it was markedly hyperintense (**B**). On the contrast-enhanced scan, the hepatic lesion presented non-rim hyperenhancement in the arterial phase (**C**), but it was hypointense in the portal venous and late phases (**D**). The capsule-like hyperenhancement area was also observed in the late phase ((**E**), arrow). According to the American College of Radiology Liver Imaging Reporting and Data System (LI-RADS), it was classified as LR-5 [1].

**Figure 3 diagnostics-13-00042-f003:**
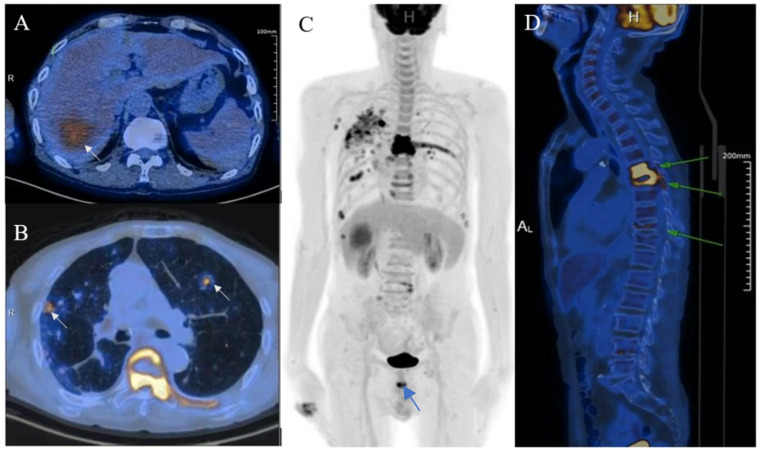
With suspicions of hepatocellular carcinoma (HCC), the patient was urged to undergo fluorine-18-fluro-2-deoxy-D-glucose-positron emission tomography/computed tomography (^18^F-FDG-PET/CT). The mass showed mild ^18^F-FDG uptake with SUVmax of 4.7 ((**A**), arrow). Moreover, hypermetabolic nodules in the lung, chest wall, thoracic vertebra, and pelvis were also observed, which indicated metastasis ((**B**–**D**), arrows).

**Figure 4 diagnostics-13-00042-f004:**
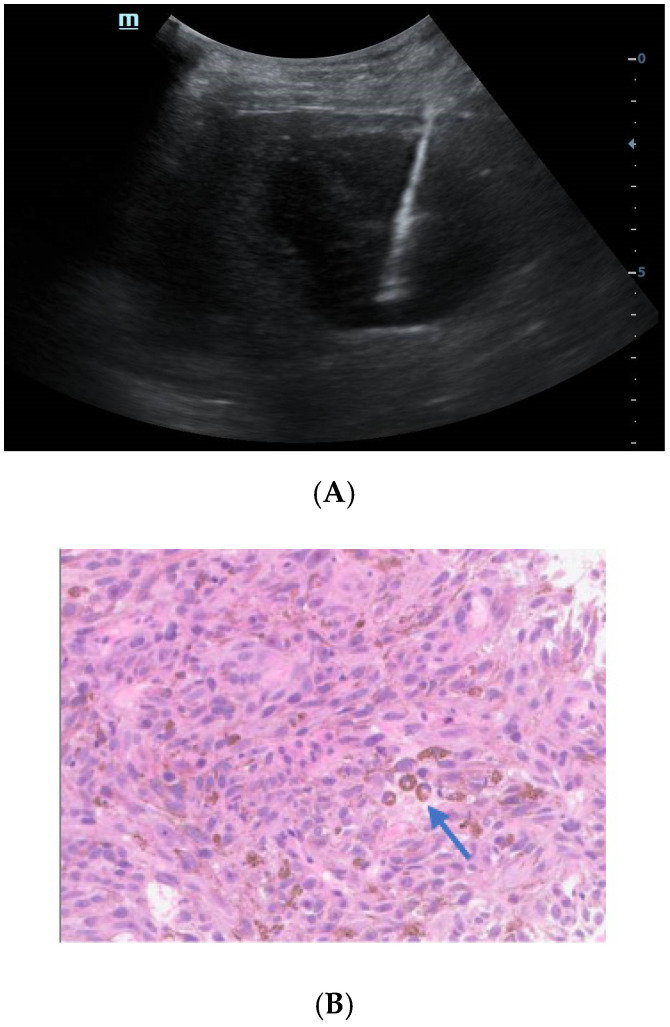
Ultrasound-guided percutaneous coarse needle biopsy (US-CNB) was carried out to acquire pathological diagnosis (**A**). The biopsy site and positioning were chosen after careful ultrasound examination and a spring-loaded 18-gauge core biopsy needle was used (MAXCORE^®^, Bard Medical Technologies Inc., Murray Hill, NJ, USA). Three cores of specimens were obtained and then sent for pathologic evaluation. Hematoxylin and eosin staining showed that the tumor was comprised of pigmented fusiform cells on microscopy scan ((**B**), 200×, arrow). Immunohistochemical staining demonstrated that the tumor cells were strongly and extensively stained with MITF and S-100, but it was negative for BRAF V600E. Consequently, the liver lesion was confirmed as metastatic melanoma. Of note, there was no history of cutaneous melanoma, and no primary tumor was revealed by systemic skin examination. In addition, a standard comprehensive examination for mucosal, ocular, and other occult sources did not reveal a definitive primary tumor. Due to the fever and pulmonary tuberculosis, the patient may not tolerate the side effects of systemic therapy. Finally, he refused systemic therapy and died from tumor progression seven months after the initial diagnosis. The incidence of melanoma increased rapidly in the United States as well as in Asian countries during the past few decades [2]. Over 90% of melanoma patients suffer from liver metastasis during the course of their disease, which generally implies poor prognosis [3]. Obtaining accurate diagnosis and clinical stage of the melanoma is important for individualized therapy and surveillance. However, isolated hepatic metastatic melanoma is difficult to recognize by imaging methods especially for those without a determined origin. In this case, the hepatic lesion was mistaken as hepatapostema using CEUS, whereas it was misdiagnosed as HCC using CEMRI. Clinically, the CEUS features of hepatapostema varies between different stages, presenting as temporal sub-segmental hyperenhancement without necrosis during the early stage, which is indistinguishable from solid tumors [4]. It is worth noting that the characteristic signal of melanoma is hyperintense on T1-weighted imaging due to the paramagnetism of melanin. Although rare, the imaging features of metastatic melanoma are varied, and may present as cystic, solid-cystic or solid nodule [5,6]. Thus, once it is detected, it’s difficult to recognize immediately. The treatment of malignant melanoma depends on the age, clinical staging, general condition, and personal preferences of the patient [7]. When ^18^F-FDG-PET/CT suggested that the tumor was in stage IV, histological examination was essential to determine the individual treatment schedule. US-CNB has been the procedure of choice in tissue diagnosis for its availability, high accuracy, and safety [8]. As such, an increasing number of biopsies has been performed in our center during the past few years. Whenever inconclusive imaging findings persist, US-CNB is performed by an experienced operator. The obtained tissue specimen is used for precise definition of specific biomarkers. In brief, the isolated hepatic malignant melanoma with undetermined origin is a rare condition and difficult to distinguish from other tumors through noninvasive imaging methods. CEUS and CEMRI play a fundamental role in the diagnosis of hepatic melanoma, and PET-CT is useful in clinical staging. For controversial results, US-CNB is essential to establish the pathological diagnosis.

## Data Availability

Not applicable.

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
