# Peer review of "Ultrasound-Guided Coarse Needle Biopsy Diagnosed Isolated Hepatic Malignant Melanoma with Undetermined Origin in TB Patient: A Case Report"

_diagnostics, 2022, doi:10.3390/diagnostics13010042_

Round 1
Reviewer 1 Report
Interesting presentation of a rare condition. The high quality of the images makes the article worthy of publication in the "interesting images" sections. However, the text could be presented in a clearer and more fluent manner, leaving only a brief description of the images in the captions.
Author Response
Dear Editor and Reviewers:
On behalf of my co-authours, we are very grateful to your giving us an opportunity to revise our manuscript. We appreciate you very much for your positive and constructive comments and suggestions on our manuscript entitled ” Ultrasound-guided coarse needle biopsy diagnosed isolated hepatic malignant melanoma with undetermined origin in TB patient: a case report”
Response to the comments of Reviewer #1
Comment No.1 Interesting presentation of a rare condition. The high quality of the images makes the article worthy of publication in the "interesting images" sections. However, the text could be presented in a clearer and more fluent manner, leaving only a brief description of the images in the captions.
Response: Thanks for your valuable comment, we have simplify the description of the images in the text.
Reviewer 2 Report
Referee of " Ultrasound-guided coarse needle biopsy diagnosed isolated hepatic malignant melanoma with undetermined origin in TB patient: a case report "
This is a meaningful and informative case report about the imaging findings of isolated hepatic malignant melanoma.
However, further review is required for paper acceptance to Diagnostics
Minor points
Line 107 etc. refers to malignant melanoma of unknown primary origin. Do you perform gastrointestinal endoscopy to search for the primary tumor?
As far as I have downloaded the paper in PDF format, it seems that the arrows, etc. in the figure are not the appropriate parts.
Is Line 46-52 Legend in Figure 1? It looks like body content unrelated to the figure.
Please delete the reference line that is not related to the lesion in Figure 3
Please add high power field of HE staining and other immunostaining images to Fig4.
Author Response
Dear Editor and Reviewers:
On behalf of my co-authours, we are very grateful to your giving us an opportunity to revise our manuscript. We appreciate you very much for your positive and constructive comments and suggestions on our manuscript entitled ” Ultrasound-guided coarse needle biopsy diagnosed isolated hepatic malignant melanoma with undetermined origin in TB patient: a case report”
Response to the comments of Reviewer #2
Comment No.1 Line 107 etc. refers to malignant melanoma of unknown primary origin. Do you perform gastrointestinal endoscopy to search for the primary tumor?
Response: Thanks for your valuable comment, we are very sorry for that we didn’t perform gastrointestinal endoscopy on the patient, but the 18F-FDG-PET/CT presented no hypermetabolic mass in gastrointestinal tract. In fact, we suggested gastrointestinal endoscopy to the patient to search for the primary tumor, however, the patient refused due to fever, tuberculosis and advanced stage of tumour.
Comment No.2 As far as I have downloaded the paper in PDF format, it seems that the arrows, etc. in the figure are not the appropriate parts.
Response: Thanks to reviewer for reminder, we marked the arrows in the appropriate parts.
Comment No.3 Is Line 46-52 Legend in Figure 1? It looks like body content unrelated to the figure.
Response: Line 46-52 described the medical history of the patient. We have added the title of Figure 1 in Line 46.
Comment No.4 Please delete the reference line that is not related to the lesion in Figure 3
Response: Thanks for reviewer. We deleted the reference line in Figure 3.
Comment No.5 Please add high power field of HE staining and other immunostaining images to Fig4.
Response: We have changed the HE staining of 200× in Fig4.